# Enhanced Feature Extraction Network Based on Acoustic Signal Feature Learning for Bearing Fault Diagnosis

**DOI:** 10.3390/s23218703

**Published:** 2023-10-25

**Authors:** Yuanqing Luo, Wenxia Lu, Shuang Kang, Xueyong Tian, Xiaoqi Kang, Feng Sun

**Affiliations:** 1School of Environmental and Chemical Engineering, Shenyang University of Technology, Shenyang 110870, China; 18842415180@163.com (Y.L.); lu_wx116@smail.sut.edu.cn (W.L.); tianxueyong@sut.edu.cn (X.T.); 2School of Mechanical and Control Engineering, Baicheng Normal University, Baicheng 137000, China; 3School of Mechanical Engineering, Shenyang University of Technology, Shenyang 110870, China; kangxiaoqi426@126.com (X.K.); sunfeng@sut.edu.cn (F.S.)

**Keywords:** rolling bearings, acoustic signal, feature extraction, fault diagnosis, convolutional neural network

## Abstract

The method of acoustic radiation signal detection not only enables contactless measurement but also provides comprehensive state information during equipment operation. This paper proposes an enhanced feature extraction network (EFEN) for fault diagnosis of rolling bearings based on acoustic signal feature learning. The EFEN network comprises four main components: the data preprocessing module, the information feature selection module (IFSM), the channel attention mechanism module (CAMM), and the convolutional neural network module (CNNM). Firstly, the one-dimensional acoustic signal is transformed into a two-dimensional grayscale image. Then, IFSM utilizes three different-sized convolution filters to process input image data and fuse and assign weights to feature information that can attenuate noise while highlighting effective fault information. Next, a channel attention mechanism module is introduced to assign weights to each channel. Finally, the convolutional neural network (CNN) fault diagnosis module is employed for accurate classification of rolling bearing faults. Experimental results demonstrate that the EFEN network achieves high accuracy in fault diagnosis and effectively detects rolling bearing faults based on acoustic signals. The proposed method achieves an accuracy of 98.52%, surpassing other methods in terms of performance. In comparative analysis of antinoise experiments, the average accuracy remains remarkably high at 96.62%, accompanied by a significantly reduced average iteration time of only 0.25 s. Furthermore, comparative analysis confirms that the proposed algorithm exhibits excellent accuracy and resistance against noise.

## 1. Introduction

The operation of mechanical equipment heavily relies on the performance of rolling bearings, which highlights their crucial role. Ensuring the safe functioning of bearings not only minimizes economic losses for enterprises but also prevents potential casualties [1,2]. The vibration signal emitted by rolling bearings typically carries valuable information regarding their operational faults, thereby offering a promising avenue for accurate fault diagnosis in mechanical research on fault diagnosis, which has extensively explored this area based on vibration signals.

In fault diagnosis based on vibration signals, He et al. [3] installed vibration sensors on mechanical equipment for detection and proposed a research method for fault diagnosis of rotating machinery. Chen et al. [4] used vibration sensors to detect the long-term health status of wind turbines and established a quantitative index of the damage degree of wind turbine bearing failure. Teng et al. [5] tested the bearing fault of a 2 MW wind turbine and demodulated the collected vibration signals, successfully proving that the cyclic spectrum correlation method can effectively identify the fault characteristics of rolling bearings. Qiao et al. [6] made a fault diagnosis of rotating machinery based on vibration signals and successfully extracted early weak fault characteristics of rolling bearings by using a stochastic resonance method. Miao et al. [7] proposed an improved adaptive variational mode decomposition method and used vibration signals to realize composite fault diagnosis of rotating machinery bearings.

Although the above methods can successfully identify the fault characteristics of bearings, they are all based on vibration signals. The fault diagnosis of vibration signals belongs to contact measurement, and the reasonable installation of a vibration sensor has certain requirements regarding its location. For some complex and precise equipment or for instances when the field sensor installation is limited, higher requirements for the sensor layout are put forward; in these cases, acoustic sensors can be used in a noncontact way to collect the acoustic radiation signal of the equipment [8,9]. Additionally, the irregular geometry of certain equipment often hinders direct sensor installation.

Acoustic radiation signals contain rich operating-status information of rolling bearings. In fault diagnosis research based on acoustic signals, some scholars have undertaken the following work: fault classification [10], fault prediction [11], and condition monitoring [12] of rotating machinery based on sound signals. And the integration of deep learning techniques [13,14,15,16] with acoustic radiation signals is specifically emphasized. 

For example, Josu’e Pacheco-Ch’errez [17] used three different supervised machine learning methods for comparative analysis to improve the fault prediction accuracy of rotating machinery. Wang et al. [18] proposed a multimodal sensor signal fusion method for acceleration signal and sound signal acquisition, which extracts features from the original vibration signal and acoustic signal, uses a CNN network to fuse them, and finally realizes the fault classification of rolling bearings. In order to solve the bearing fault problem of CNC machine tools, Mohmad lqbal [19] proposed a fault diagnosis method based on the acoustic signal of a convolutional neural network, and the research shows that this method can realize the classification of bearing faults. Eugenio Brusa [20] applied the transfer learning strategy to the fault diagnosis of rolling bearings and verified that the deep learning architecture based on sound signals can realize the fault diagnosis of machines. Bai et al. [21] fused sound and vibration signals to improve the detection accuracy of rolling bearing fault characteristics, which is conducive to the condition monitoring of bearing systems. Zhang et al. [22] focused on the fault diagnosis of offshore wind power equipment using acoustic emission signals and vibration signals. The advantages of acoustic emission sensors include their high sensitivity, precision, and ability to acquire large amounts of data. However, they are limited in their application to specific fields for nondestructive testing and lack wide-ranging applicability.

Although the aforementioned studies made certain advancements in deep learning, their utilization of convolution kernel size is overly simplistic and predominantly relies on single-channel diagnosis, thereby limiting the effective extraction of more comprehensive fault feature information. Moreover, the sound signal is seriously disturbed by background noise, so the question of how to filter the collected sound signal is very important. For complex equipment in the process of operation, the collected signal contains a lot of interference information and shows strong nonstationarity. If the collected original signal is directly input into the neural network, the network will learn many invalid features, resulting in a reduction in classification accuracy. Therefore, the question of how to effectively use sound signal and a deep learning algorithm to achieve fault diagnosis of rolling bearings is very important.

Based on the above research, this paper presents a new method of rolling bearing fault diagnosis based on a multichannel acoustic array signal and multinetwork module combination. The main work of this paper is as follows:(1)An IFSM is developed, which utilizes three convolutional filters of varying sizes to process input image data.(2)The CAMM is constructed, and it is utilized to assign weights to all branch channels, thereby achieving the refinement of fault information.(3)The research on fault diagnosis method for rolling bearings based on sound signals is accomplished by constructing a deep learning network framework that integrates the IFSM, CAMM, and CNNM.

The rest of this article is described below. In Section 2, the proposed theoretical method is introduced in detail. In Section 3, experiments are used to verify the effectiveness of the proposed method. Finally, the conclusion of this paper is introduced.

## 2. Enhanced Feature Extraction Network

### 2.1. Multichannel Acoustic Array Data

Compared with the single microphone sensor, the microphone array adopts multiple acoustic sensors to collect data on the running state of the rolling bearing. The ring array can collect the characteristics of the bearing’s circumferential and radial running state. The schematic diagram of the acoustic array measurement points is shown in Figure 1. The fault characteristic information of different bearing types can be collected from multiple angles by installing multiple acoustic sensors on the ring disc to improve the fault diagnosis and identification accuracy of rolling bearings.

Multiple ring array sensors were used for data acquisition, and each data acquisition channel was arranged in parallel to construct a two-dimensional spatial data matrix, and each independent channel also contained rich fault feature information of rolling bearings. In order to extract effective fault feature information at a deeper level, this paper proposed an information feature selection module. And because each channel is both independent and interrelated, it is therefore very important to assign weight to the fault information contained in each channel. This paper uses the channel attention mechanism module to achieve this function.

### 2.2. Data Preprocessing

Since the two-dimensional image signal contains higher-dimensional fault feature information, this paper converts the collected one-dimensional sound signal into a two-dimensional gray image for feature input. Two-dimensional images provide more abundant fault feature information, which can improve the sample quality of model training.

Firstly, the collected sound signals are normalized. The value range of the normalized data is 1 to 0, corresponding to the change in brightness and darkness of the gray value in the grayscale image, and its mathematical expression is as follows:(1)$$x=\frac{x-x_{min}}{x_{max}-x_{min}}$$
where *x* represents the input signal, and *x_min_* and *x_max_* represent the minimum and maximum values of signal *x*, respectively. The output of an RGB image does not generate redundant features; thus, gray images convert the three channels into one channel without any additional operations, resulting in a significant reduction in computational requirements. Then, the collected one-dimensional data signal is reconstructed to form a gray image. In order to optimize the efficiency of network training, it is advisable to limit the size of the fully connected layer structure. This study employs a sampling point length of 1024 for the intercepted one-dimensional signal. Additionally, for computational convenience, the input gray image is designed with equal length and height. Consequently, the one-dimensional signal is evenly divided into 32 segments on average, each segment having a length of 32. Every 32 data points form a column of the grayscale map, and 32 segments of data are stacked with 32 columns and finally reconstructed into 32 × 32 two-dimensional grayscale images. The detailed operation is shown in Figure 2.

### 2.3. IFSM

The utilization of smaller convolution kernels enables the extraction of more localized features, whereas larger convolution kernels facilitate the extraction of more global features. Generally, incorporating multiple convolution cores of varying sizes can enhance network performance [23,24]. To enhance fault information feature extraction capability, three distinct convolution kernels are selected. The IFSM is shown in Figure 3. Three convolution kernels of different scales are used to carry out convolution operations, and the results of the operations are spliced to obtain signals:(2)$$x_{c}^{conv}=concat(conv_{1}(x_{c})+conv_{2}(x_{c})+conv_{3}(x_{c}))$$
where *x_c_* indicates the *L* × *L* dimension number *x* of *c* channels. *conv*_1_, *conv*_2_, and *conv*_3_ represent convolution operations with convolution kernel sizes of 3 × 3, 5 × 5, and 7 × 7, respectively. And *concat* represents the feature concatenation operation. Then, $x_{c}^{conv}$ is then fused by a convolution kernel of size 1 × 1 to obtain $x_{}^{conv}$. Finally, the feature weight of input signal *x* is obtained by sigmoid activation function:(3)$$\omega_{c}^{}=\sigma \left(conv_{4}(x_{}^{conv})\right)$$
where $\sigma (x)=1/(1+e^{-x})$ represents sigmoid activation function, and *conv*_4_ stands for convolution. Finally, after processing by IFSM module, output signal $x_{c}^{IFSM}$ is obtained:(4)$$x_{c}^{IFSM}=x_{c}\times \omega_{c}$$

### 2.4. CAMM

Considering that different channels contain different contribution degrees of fault-characteristic information, the channel attention mechanism is used to assign different weight values to each channel [25,26]. The specific operation process is shown in Figure 4. Firstly, the input signals $x_{c}^{IFSM}$ are processed by global average pooling and global maximum pooling, respectively. The acoustic signals of each channel are compressed and the fault features of each channel are compressed into a global feature. Then, the generated two feature maps are fed into the multilayer perceptron (MLP) with shared weights for interchannel learning, and the dimensionality between the two neural layers is reduced by compression ratio. Finally, the MLP output features are added and activated by sigmoid function to generate channel weight *ω*_c_ and calculate the output features according to element multiplication. The expression is defined as follows:(5)$$\omega_{c}^{}(x_{c}^{IFSM})=\sigma \left(MLP\left(AvgPool(x_{c}^{IFSM})\right)+MLP(MaxPool(x_{c}^{IFSM}))\right)$$
(6)$$x_{c}^{CAMM}=\omega_{c}\times x_{c}^{IFSM}$$
where $\sigma \left(\cdot \right)$ represents sigmoid activation function, and $x_{c}^{IFSM}$ indicates the signal processed by IFSM. $x_{c}^{CAMM}$ is the output signal after CAMM. *AvgPool* and *MaxPool* represent average and maximum pooling operations, respectively.

### 2.5. CNNM

The fault diagnosis of rolling bearings is achieved through the CNNM network after successfully completing IFSM and CAMM. The classical structure of CNN is shown in Figure 5, which is mainly composed of input layer, convolution layer, pooling layer, activation function layer, and full connection layer.

(1)Convolutional layer.

The 2D convolution operation is defined as follows:(7)$$g_{2}(l)=\omega_{i,j}^{l}\sum_{i=1}^{m}\sum_{j=1}^{n}x_{i,j}^{l}+b^{l},l=1,2,\ldots ,L.$$
where $g_{2}(l)$ represents the features extracted from the lth convolution kernel; $\omega_{i,j}^{l}$ represents the weight coefficient; *b*^l^ stands for the deviation coefficient; and *m* and *n* indicate the size of the input information.

(2)Pooling layer.

After the convolution operation, the linear rectification function (ReLU) is used to carry out nonlinear transformation of the obtained data results. The formula is as follows:(8)$$x_{i,j}^{l+1}=f(g_{2}(l))=\max\{0,g_{2}(l)\},l=1,2,\ldots ,L.$$

The pooling layer is equivalent to downsampling, which compresses the input information features, thus speeding up the operation speed of the neural network. This paper adopts the maximum pooling algorithm, which is defined as follows:(9)$$p_{i,j}(l)=\max\{x_{i,j}(lN,(l+1)N)\},l=1,2,\ldots ,L.$$
where *N* represents the size of the convolution kernel and *l* represents the *l*th pooling kernel.

(3)Fully connected layer.

The fully connected layer integrates the features extracted by the previous layer network and maps these features to the sample label space. The fully connected layer weights and sums the output features of the previous layer and inputs the results into the activation function to complete the classification of the target.
(10)$$y_{i}^{pre}=f\left(\mathrm{ReLU}\left(\omega^{fc}x_{i,j}+b\right)\right)$$
where $w^{fc}$ is the weight of the fully connected layer and $f\left(\cdot \right)$ is the softmax activation function.

(4)Parameter configuration.

In this study, EFENet is an end-to-end model that relies on a backpropagation algorithm to update parameters during training. The loss function of EFENet is the cross-entropy loss function:(11)$$Loss=-\sum_{i}y_{i}\log(y_{i}^{pre})$$
where $y_{i}$ is the actual label and $y_{i}^{pre}$ is the predicted label.

The present study proposes a feature-enhanced deep learning network and utilizes acoustic array signals for the purpose of fault diagnosis in rolling bearings, based on the aforementioned research foundation. The overall logic block diagram is illustrated in Figure 6, with the specific steps outlined as follows:Step 1:acquire multichannel data of rolling bearings using an acoustic array sensor;Step 2:convert one-dimensional data into a two-dimensional grayscale image;Step 3:feed the 2D grayscale images generated by each channel into the IFSM module for deep learning;Step 4:apply the channel attention mechanism module to weight all channels;Step 5:input the fused data into the CNNM module for fault feature extraction;Step 6:obtain classification results as output.

## 3. Case Studies

### 3.1. Experiment Introduction

In this study, in order to verify the feasibility of the proposed method, experimental data were obtained from the rolling bearing test bench. The general technical route is shown in Figure 6. The experimental test system included the motor, controller, rotating shaft, experimental bearing, acoustic array sensor, and data collector, as shown in Figure 7. Sixteen acoustic sensors are mounted on a circular disk in the form of a ring array. The sensor model number is BSWA MPA416, and the sensor sensitivity is 50 mV/Pa; according to the sampling theorem, the sampling frequency of each channel is set to 16,384 Hz. the attenuation of the signal becomes more pronounced when it is positioned 500 mm away from the transmitting source. To facilitate the acquisition of the acoustic array signal, the acoustic array sensor is installed on the plane 200 mm away from the bearing under test; the center point of the sensor is in a straight line with the center of the bearing under test. The status information of the bearing during operation is collected by the acoustic array sensor, and the collected acoustic signals are transmitted to the data acquisition system. The acoustic array sensor is shown in Figure 8a. The model of the data collector is PAK MKII-SC42, as shown in Figure 8b.

The running state of rolling bearings mainly includes the following: normal, inner ring failure, outer ring failure, rolling element failure, and cage failure. Bearing assemblies with different faults were tested on the experimental bench. The geometric parameters of the experimental bearings are shown in Table 1.

The different failure components are shown in Figure 9.

In the process of dataset processing, the failures can be categorized into seven types, which include inner failure, outer failure, rolling element failure, cage failure, and coupling failure. Each fault type contains 500 samples, of which 80% of the dataset is used for network training, while the remaining 20% of the dataset is used for testing and verification. The details of the dataset are shown in Table 2. The fault types are categorized with numerical labels ranging from C0 to C6. In addition, the tenfold crossover method was used for comparative analysis in the experiments. The length of the data sample is 1024. The motor speed is 1200 rpm. The judicious selection of batch size and learning rate not only yields a relatively robust model but also significantly reduces computational overhead. The model optimizer utilizes the Adam algorithm with a learning rate of 0.001 and epochs set to 100. Both training and testing sample batch sizes are set to 36, while model parameters are updated automatically through backpropagation. Taking measuring point 1 as an example, the time-domain signals of seven different fault types are shown in Figure 10.

In the process of neural network training, in order to improve the generalization ability of the network, we use batch normalization (BN) technology. BN is usually placed between the convolutional layer and the pooling layer. In order to prevent overfitting during training, the dropout layer is introduced, and the value of dropout in this paper is set to 0.5. The computer is configured as follows: 11th Gen Intel(R) Core(TM) i7-11700K @ 3.60 GHz, RAM 48 G, NVIDIA GeForce GTX 3070 GPU and made in Texas, USA. EFENet is built under Torch-gpu 1.11.0 based on Python 3.10 [27]. Table 3 shows the detailed information of the proposed algorithm framework.

### 3.2. Analysis Results

The training and testing process and loss value of the proposed algorithm are shown in Figure 11. It can be seen from Figure 11 that the training accuracy of the proposed algorithm reaches convergence when it iterates about 18 times. The final accuracy of the test set is maintained at about 98%. The loss value of the model decreases rapidly with the increase in the number of iterations and finally remains stable. From the analysis results, it can be concluded that the method proposed in this paper can effectively propose the fault characteristics of rolling bearings from the acoustic array signal. In order to further analyze the difficulty of feature extraction of each type of sound fault signal, the confusion matrix of the test set is shown in Figure 12. The classification accuracy of C0 and C6 fault data is poor, but it is also maintained at 95%. The diagnostic accuracy of other categories is above 98%, which indicates that the proposed method can effectively diagnose the running state of rolling bearings under different fault conditions.

Figure 13 describes the weight proportion of seven types of faults under different channels. It can be clearly seen from Figure 10 that different acoustic array channels have different weight contribution values. Channel 3 and channel 14 have the largest weight contribution values, with a weight coefficient of 0.8, while the weight contribution values of other channels are relatively low. It is further proved that it is necessary to apply the channel attention mechanism module.

In order to understand the details of data processing in each process of the model, a t-distributed stochastic neighbor embedding algorithm (t-SNE) is used to process the initial input stage, information feature selection, channel attention mechanism, and convolutional neural network of various fault data. The results are shown in Figure 14. It can be seen from Figure 14a that when the data first entered the model, the distribution was irregular and random. In Figure 14b, the feature distribution gradually became regular, and data of different categories began to gather within the class. In Figure 14c, the spacing within the class gradually increased to achieve effective classification results. The classification effects of various faults are clearly discernible in Figure 14d, ultimately.

### 3.3. Comparison of Results with Other Methods

In order to verify the fault diagnosis performance of the proposed algorithm, the proposed method is compared with a residual convolutional autoencoder (RCAE) [23], sparse autoencoders (SAE), CNN, DenseNet, and ResNet. Detailed parameters of the other five methods are shown in Table 4. The classifier structure is the same as the EFENet network structure “2048-1024-256-7”. The RCAE architecture comprises two conv-pool layers and one deconv layer. The CNN network is composed of three conv-pool layers. The ResNet and DenseNet networks consist of a residual module and a dense connection block module (den), respectively. The convolutional kernel has a size of 3 and a stride of 1, while the pooling operation utilizes maximum pooling with a stride of 2. The batch size is set to 36 and the learning rate to 0.005. The comparison of the results of five different methods’ operations are shown in Figure 15. It is evident from the figure that each method exhibits an average classification accuracy of 98.52%, 97.36%, 93.58%, 82.36%, and 96.42%, respectively, with the SAE method demonstrating the lowest diagnostic accuracy. Notably, our proposed method surpasses CNN by a margin of 7% and ResNet by a margin of 4%. These comparative findings highlight the superior fault diagnosis performance achieved by our proposed approach, further validating the effectiveness of the information feature selection module and channel attention mechanism module in extracting bearing fault features from acoustic array signals.

The proposed method was further validated by employing sensitivity, specificity, precision recall, and F1-score to analyze the specificity results in comparison with other methods. The formula for each indicator is defined as follows:Sensitivity = TP/(TP + FP)(12)
Specificity = TN/(TN + FN)(13)
Precision = TP/(TP + FP)(14)
Recall = TP/(TP + FN)(15)
F1-Score = 2 × (Precision × Recall)/(Precision + Recall)(16)
where TP represents true positive, which refers to the number of correctly predicted positive samples. FP represents false positive, indicating the number of incorrectly predicted positive samples. TN stands for true negative, denoting the number of accurately predicted negative samples. FN signifies false negative, representing the number of erroneously predicted negative samples. The comparison results of the five methods are shown in Table 5. The values of each index of the proposed method, as shown in Table 5, surpass those of other methods, thereby further substantiating the accuracy of the proposed method.

### 3.4. Antinoise Experimental Analysis

In order to further evaluate the noise absorption capability of the algorithm proposed in this paper, 2 dB white Gaussian noise was added to the collected acoustic array signal sample, and the SNR formula is defined as follows:SNR = 10lg(*P_signal_*/*P_noise_*)(17)

The three methods, namely, RCAE, DenseNet, and ResNet, which exhibited relatively good diagnostic performance in the previous section were selected and compared with the approaches proposed in this paper. The comparative results are presented in Table 6. These findings demonstrate that EFENet achieves a fault diagnosis accuracy as high as 96.62%, greater than the other three methods and indicating its excellent antinoise capability and superior robustness. Furthermore, the average iteration time of the EFENet model is recorded at 0.25 s, highlighting its computational efficiency suitable for acoustic-signal-based rolling bearing fault diagnosis applications.

The t-SNE clustering results of the four methods are presented in Figure 16. It is evident from the figure that the fault clustering performance of the ResNet method is subpar, as all seven types of fault data appear to be intertwined without effective separation. While both the DenseNet and RCAE methods exhibit a relatively satisfactory ability to separate most of the fault data, there still exist instances of misclassification and indistinct boundaries. In contrast, our proposed method demonstrates superior classification effectiveness with distinct boundaries and substantial interclass spacing, further substantiating its superiority.

The accuracy curve during the training process of the four test sets is depicted in Figure 17. It can be observed from the figure that the RCAE method exhibits the most rapid convergence speed. However, this method’s accuracy is unstable, averaging 95%. The ResNet and DenseNet methods demonstrate relatively lower accuracies, reaching approximately 94% and 91%, respectively. In comparison to these approaches, our proposed method showcases a relatively fast convergence speed with a peak accuracy rate of 97%. Furthermore, our proposed method demonstrates stable convergence accuracy, providing further evidence for its ability to suppress noise and achieve high robustness under noisy conditions.

The Equations (12)–(16) indicators were retained for the analysis of all methods, and the resulting findings are presented in Table 7. As can be seen from Table 7, the values of each index of the proposed method are greater than those of the other methods, which once again proves that the proposed method has good antinoise ability.

## 4. Conclusions

The installation of vibration sensors may not be feasible under certain special conditions, leading to serious problems with susceptibility to background noise interference in sound signals. To address this issue, a research method for rolling bearings is proposed that includes data preprocessing, information feature selection, a channel attention mechanism, and a convolutional neural network. The following conclusions are drawn:(1)The EFENet network effectively proposes fault characteristics of rolling bearings from acoustic array signals. Furthermore, this method exhibits fast convergence speed and maintains a test set accuracy of 98%.(2)Compared with RCAE, CNN, SAE, DenseNet, and ResNet, it can be observed that the fault classification accuracy of the EFENet network reaches as high as 98.52%, which is 7% higher than CNN and 4% higher than ResNet. The proposed method in this paper achieves superior fault diagnosis performance. It further demonstrates that the information feature selection module and channel attention mechanism module proposed in this study effectively extract bearing fault feature information from acoustic array signals.(3)During the antinoise experiment process, t-SNE results of the EFENet network exhibited clear boundaries and significant spacing between classes. This indicates that EFENet accuracy and stability hold even under noisy background conditions. The algorithm proposed in this paper holds certain engineering application value for rolling bearing fault diagnosis.

The method proposed in this paper offers the advantages of convenient sensor installation, high algorithm accuracy, and fast calculation speed. It holds significant practical value for engineering applications. Aiming at the computational efficiency and overfitting problems of the model, we will take into consideration novel models for lightweight pretrained deep learning models in future endeavors. The future research endeavors of our team will focus on the development and refinement of fault life prediction methods for rolling bearings, specifically utilizing acoustic radiation signals.

## Figures and Tables

**Figure 1 sensors-23-08703-f001:**
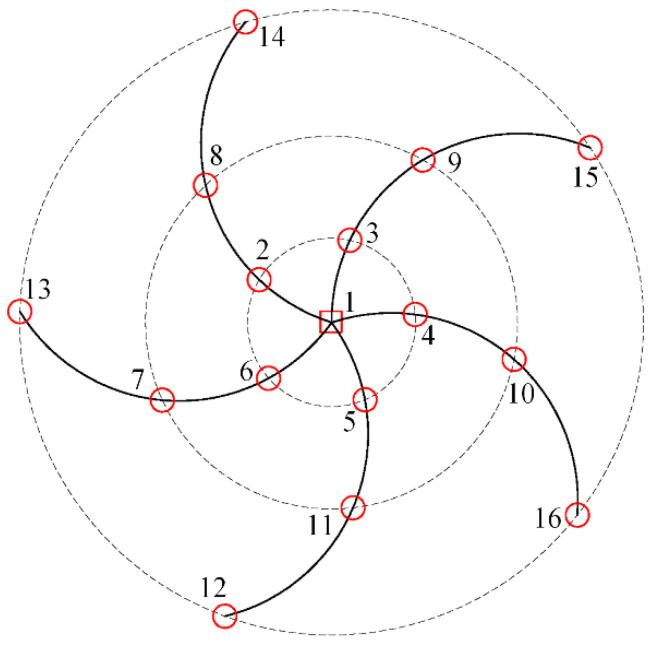
Schematic diagram of acoustic array measuring points.

**Figure 2 sensors-23-08703-f002:**
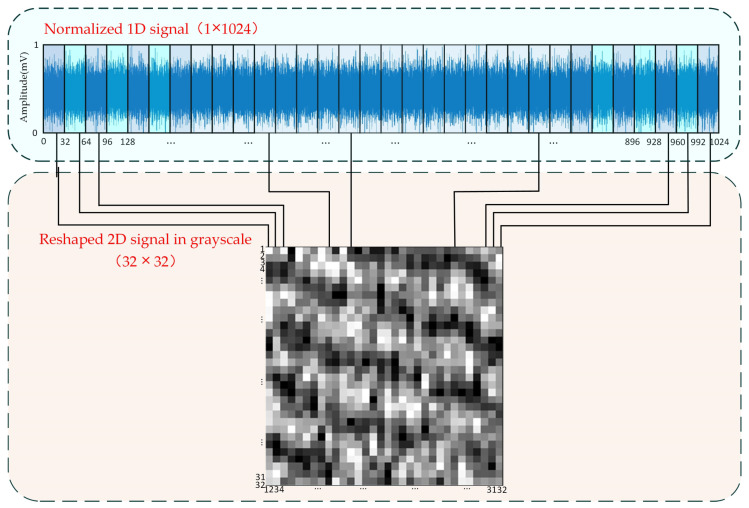
Two-dimensional grayscale map after reshaping process of acoustical signal.

**Figure 3 sensors-23-08703-f003:**
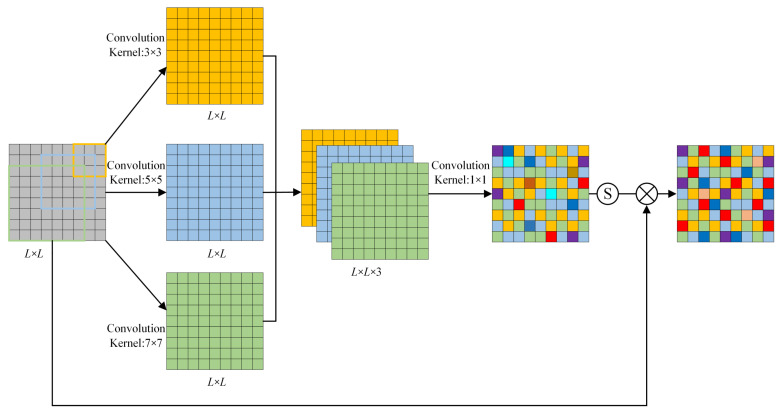
Information feature selection module.

**Figure 4 sensors-23-08703-f004:**
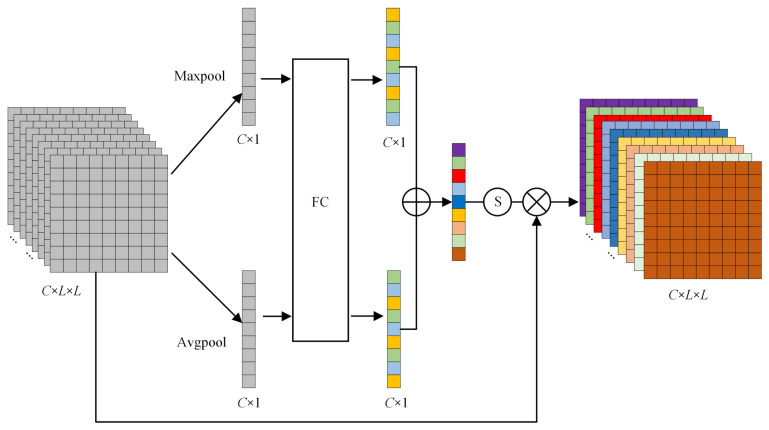
Channel attention mechanism module.

**Figure 5 sensors-23-08703-f005:**
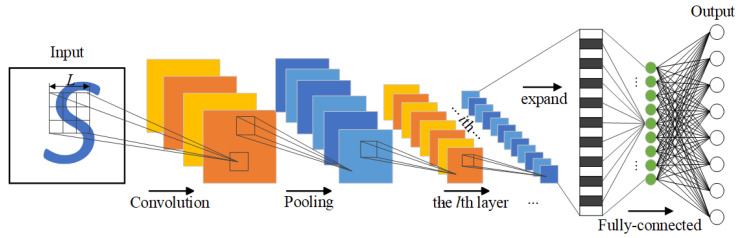
Convolutional neural network structure diagram.

**Figure 6 sensors-23-08703-f006:**
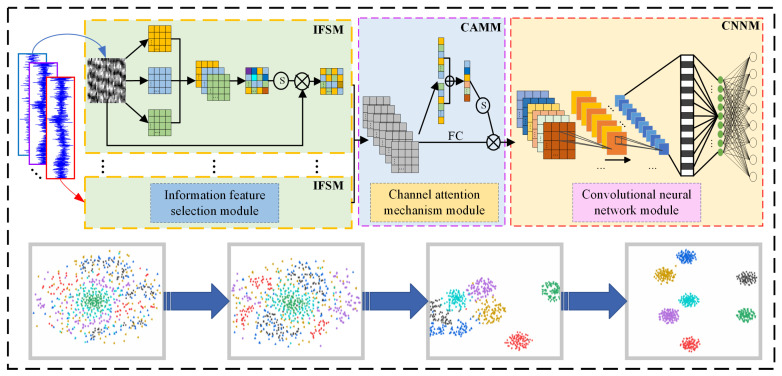
EFENet-based bearing fault diagnosis.

**Figure 7 sensors-23-08703-f007:**
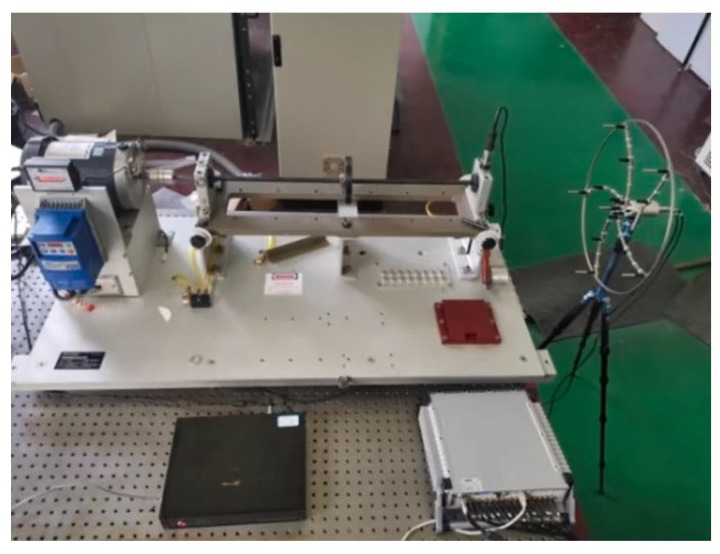
The experimental test device.

**Figure 8 sensors-23-08703-f008:**
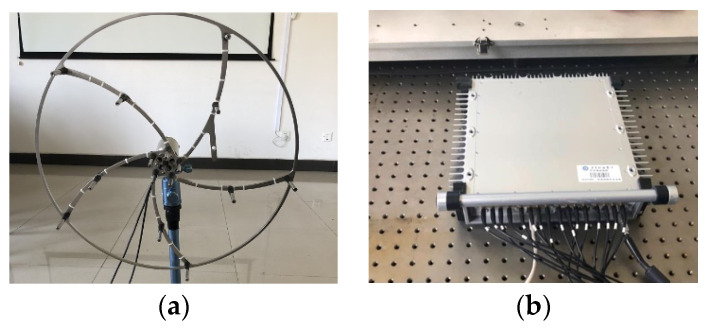
Test equipment: (**a**) acoustic array sensor; (**b**) data collector.

**Figure 9 sensors-23-08703-f009:**
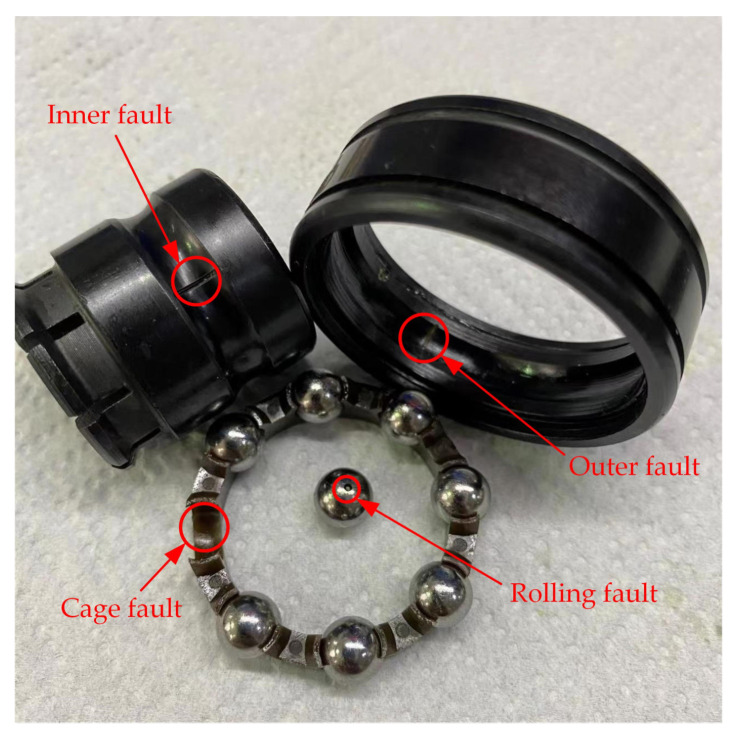
The different failure components.

**Figure 10 sensors-23-08703-f010:**
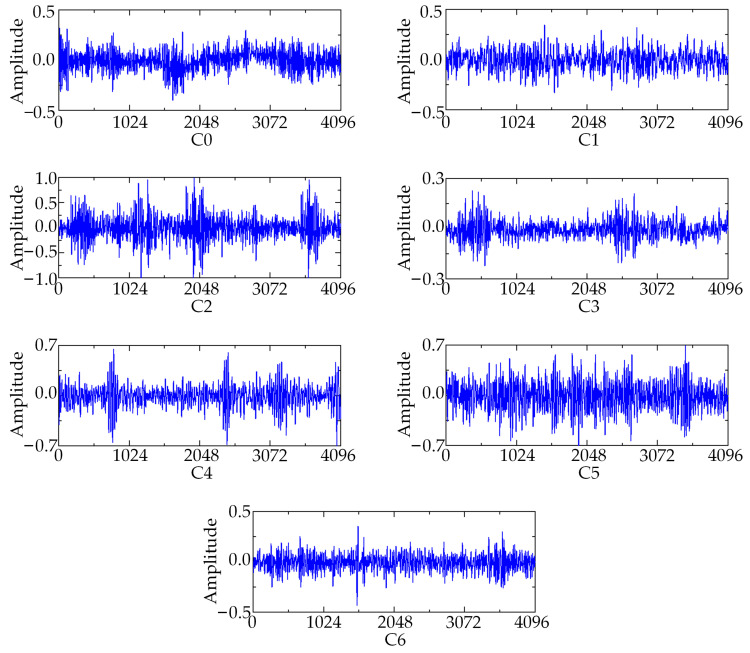
The time-domain signals of 7 different fault types.

**Figure 11 sensors-23-08703-f011:**
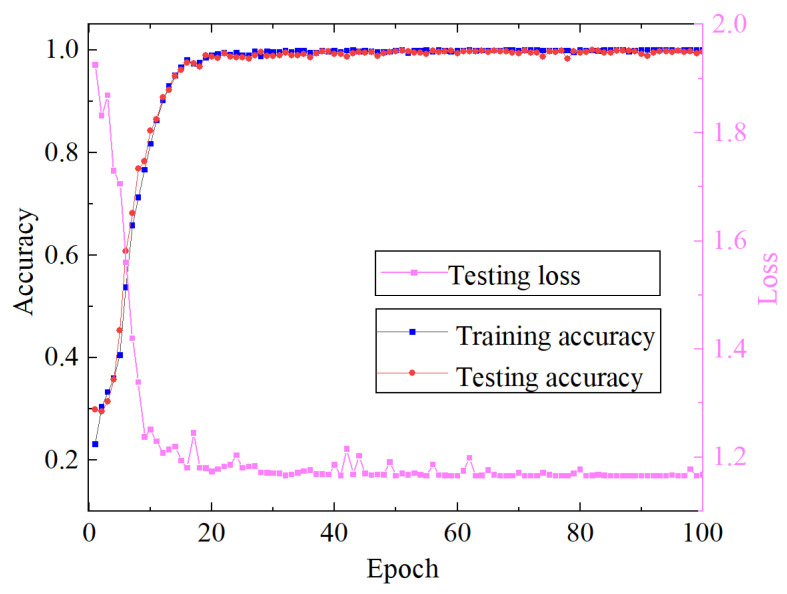
Loss and accuracy of the EFENet training process.

**Figure 12 sensors-23-08703-f012:**
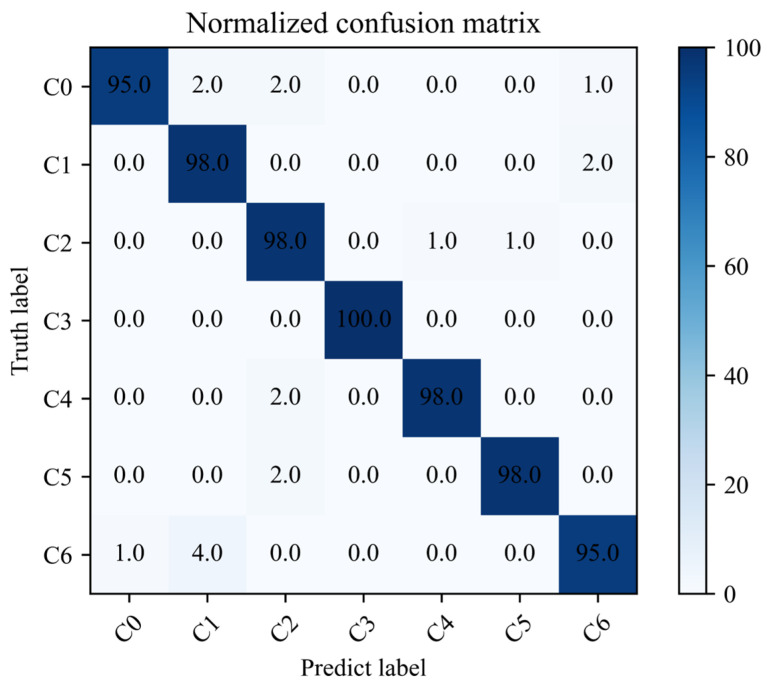
The confusion matrix results for EFENet.

**Figure 13 sensors-23-08703-f013:**
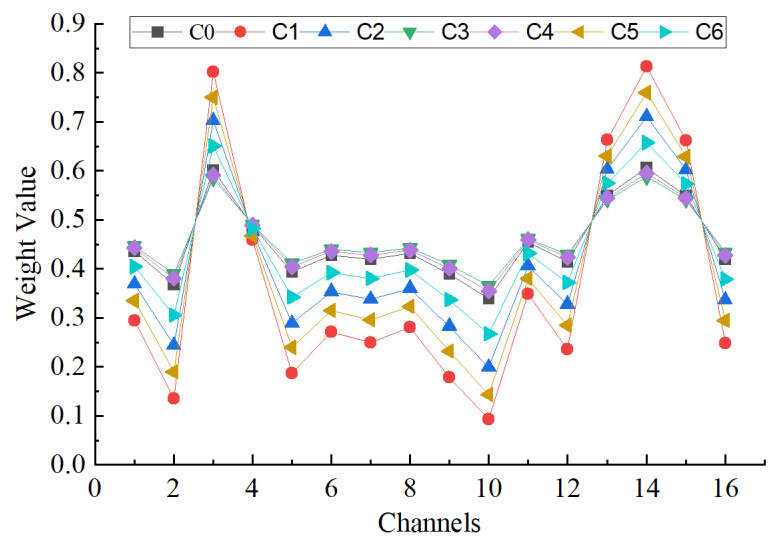
The weight values of different faults on different channels.

**Figure 14 sensors-23-08703-f014:**
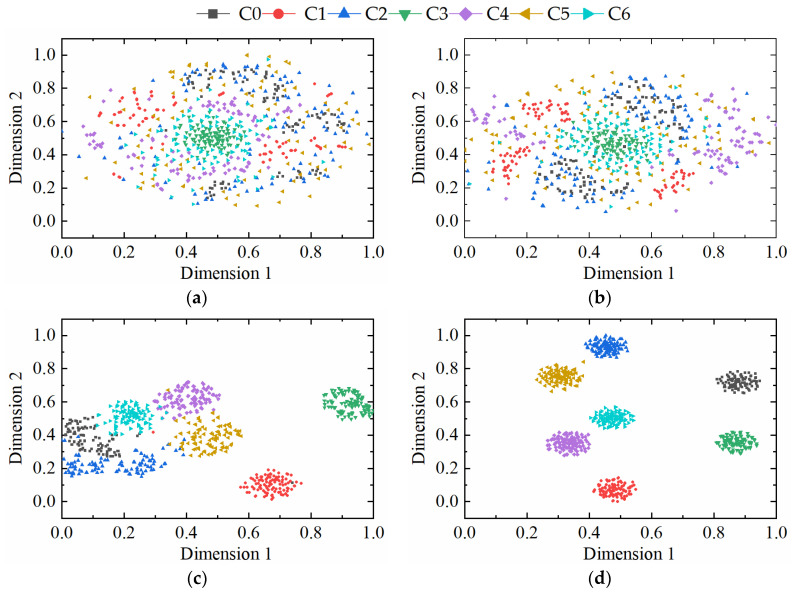
Visualization results of EFENet method:(**a**) Original input. (**b**) IFSM-CFSM. (**c**) 2D-CNN. (**d**) Classifier.

**Figure 15 sensors-23-08703-f015:**
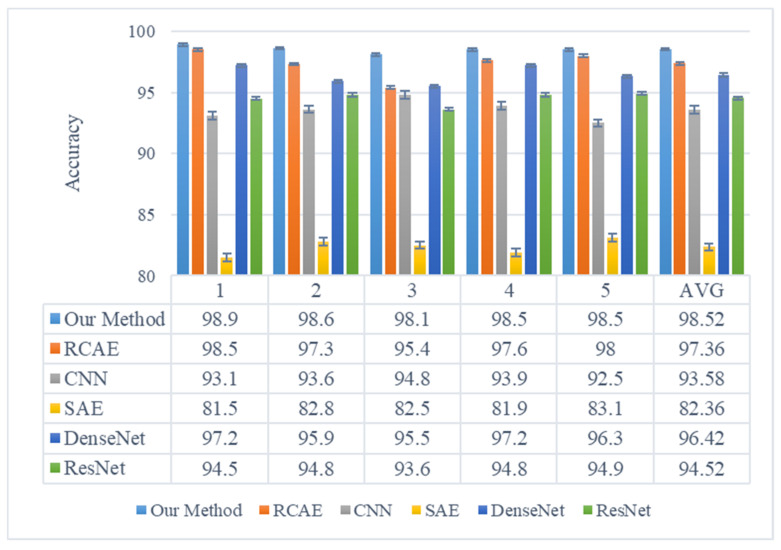
The results of different methods.

**Figure 16 sensors-23-08703-f016:**
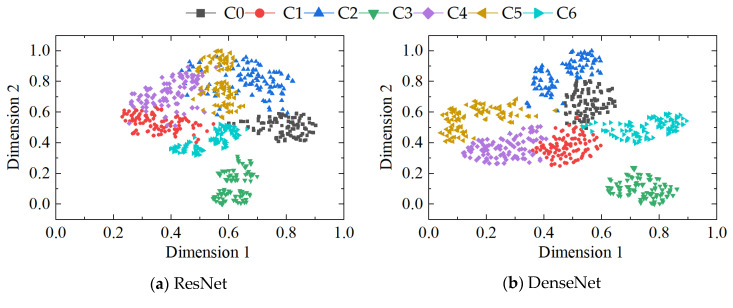
Visualization of features from (**a**) ResNet, (**b**) DenseNet, (**c**) RCAE, and (**d**) our method.

**Figure 17 sensors-23-08703-f017:**
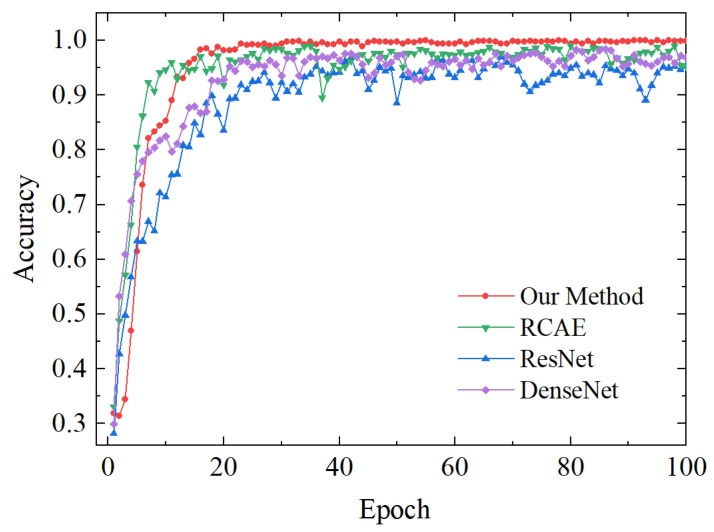
Accuracy of four methods during training process.

**Table 1 sensors-23-08703-t001:** Structural parameters of rolling bearings.

Structural Parameters	Parameter Values	Structural Parameters	Parameter Values
Bearing type	MB ER-8K	Contact angle	0°
Inside diameter	0.91 in	The number of rollers	8
Pitch diameter	1.32 in	Roller diameter	0.312 in

**Table 2 sensors-23-08703-t002:** Setup of the fault dataset of bearings.

Bearing Condition	Working Speed (r/min)	Training/Testing Sample Size	Label
Two normal bearings	1200	400/100	C0
Outer race failure (near the sensor), one normal	1200	400/100	C1
Inner race failure (near the sensor), one normal	1200	400/100	C2
Ball failure (near the sensor), one normal	1200	400/100	C3
Compound failure of inner and outer races (near the sensor), one normal	1200	400/100	C4
Outer race failure (near the sensor), inner race failure	1200	400/100	C5
Outer race failure, inner race failure (near the sensor)	1200	400/100	C6

**Table 3 sensors-23-08703-t003:** The structure and parameter setting of the method are presented.

Type of Layer	Parameters	Output Size
Input layer	-	[16@1024 × 1]
Image conversion		[16@32 × 32]
IFSM		
Conv layer 1	Kernel = [3 × 3], stride = 1	[16@32 × 32]
Conv layer 2	Kernel = [5 × 5], stride = 1	[16@32 × 32]
Conv layer 3	Kernel = [7 × 7], stride = 1	[16@32 × 32]
Concat	Kernel = [1 × 1], stride = 1	[16@32 × 32]
CFSM		
Avg-pool	Kernel = [32 × 32], stride = 1	-
Max-pool	Kernel = [32 × 32], stride = 1	-
2D-CNN		
Conv layer 1	Kernel = [7 × 7], stride = 2	[16@32 × 32]
Pooling layer 1	Kernel = [2 × 2], stride = 2	[16@16 × 16]
Conv layer 2	Kernel = [2 × 2], stride = 1	[32@16 × 16]
Pooling layer 2	Kernel = [2 × 2], stride = 2	[32@8 × 8]
Classifier	2048-1024-256-7	7

**Table 4 sensors-23-08703-t004:** Structure and parameter setting of comparison methods.

Models	Parameters
RCAE [25]	2*[Conv-Pool]-1*[Deconv]-Classifier
CNN	3*(Conv-Pool)-Classifier
ResNet	2*[Conv-Pool]-1*Res-Pool-Classifier
DenseNet	2*[Conv-Pool]-1*Den-Pool-Classifier
SAE	2048-1024-256-7

**Table 5 sensors-23-08703-t005:** Comparative results of different evaluation indexes.

	Sensitivity	Specificity	Precision	Recall	F1-Score
Our method	0.98	0.99	0.97	0.97	0.97
RCAE	0.97	0.97	0.95	0.96	0.95
CNN	0.93	0.94	0.92	0.93	0.92
SAE	0.82	0.83	0.79	0.81	0.80
DenseNet	0.96	0.95	0.91	0.94	0.92
ResNet	0.95	0.94	0.93	0.93	0.93

**Table 6 sensors-23-08703-t006:** Results of four models based on fivefold cross-validation.

	1st	2nd	3rd	4th	5th	Average	Average Iteration Time
ResNet	88.7	88.3	90.2	89.6	88.9	89.14	3.61 s
DenseNet	92.6	92.3	91.9	90.8	91.6	91.84	17.49 s
RCAE	94.2	94.8	93.2	93.8	94.5	94.10	0.32 s
Our method	96.8	96.3	96.7	97.1	96.2	96.62	0.25 s

**Table 7 sensors-23-08703-t007:** Comparative results of different evaluation indexes after introducing noise.

	Sensitivity	Specificity	Precision	Recall	F1-Score
Our method	0.96	0.96	0.96	0.97	0.96
RCAE	0.93	0.95	0.95	0.94	0.94
DenseNet	0.90	0.91	0.89	0.90	0.89
ResNet	0.88	0.89	0.87	0.88	0.87

## Data Availability

The detailed data supporting the results of this study are available from the corresponding authors upon request.

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
