# Peer review of "Enhanced Feature Extraction Network Based on Acoustic Signal Feature Learning for Bearing Fault Diagnosis"

_sensors, 2023, doi:10.3390/s23218703_

Round 1
Reviewer 1 Report
This paper proposed an enhanced feature extraction network model based on acoustic signal feature learning. This method not only enables non-contact measurement but also provides comprehensive state information during equipment operation. Overall, the paper is interesting and the structure is reasonable. However, there exist some issues that need to be improved, which are listed in the following.
1. The "Abstract" sections can be made much more impressive by highlighting your contributions. Contributions 1 and 3 cannot be considered as primary contributions and need to be reconsidered.
2. In the information feature selection module, Why are three different convolutional kernels designed? Please provide a detailed explanation of the reason.
3. The convolution process shown in Figure 5 is already familiar to us, and it is recommended to reduce the description of this section.
4. It is recommended to mark the failure positions of different fault modes in Figure 9.
5. What is the difference between acoustic emission signals and acoustic signals? Such as, integrated intelligent fault diagnosis approach of offshore wind turbine bearing based on information stream fusion and semi-supervised learning.
6. In section 2, only a theoretical description of the method is provided. What is the overall process for the proposed method? What is the training process of the model? These issues need to be explained in detail.
7. Whether the proposed method can realize real-time prediction.
no
Reviewer 2 Report
The subject addressed is current and has interest.
The work is well structured and clearly laid out.
The methodology and the presentation of results should be improved.
The conclusions are supported in the presented results.
I consider there are some aspects that should be corrected/better described:
- In lines 87 and 88, the paragraph ends in a descriptive way (with a colon), so the next paragraph must be linked to the previous one by marks, corresponding to the description;
- In Figure 2, the amplitude referred in Normalized 1D signal have m/s2 units, but as the measure correspond to a sound pressure, shouldn't the units be in Pa or dB or even in volts?
- I consider that the operations performed in the 2.3 and 2.4 subsections should be better reasoned or based in literature;
- In line 315 is referred the Figure 14a, but in Figure 14 the four presented graphs are not labeled;
- In lines 316 to 319, the different stages described in this text should be identified in figure 14, whose four graphs should be properly labeled (a, b, c. and d).
Reviewer 3 Report
Please find attached my review report.

Minor corrections are needed regarding the quality of English language.
Reviewer 4 Report
This paper proposes an enhanced feature extraction network (EFENet) for fault diagnosis of rolling bearings, which is based on acoustic signal feature learning. The EFENet consists of four main components: a data preprocessing module, an information feature selection module, a channel attention mechanism module, and a convolutional neural network module. Generally, the topic is interesting. However, some comments need to be addressed before acceptance.
The abstract Please add the numerical findings
Introduction:
There are several modalities for detecting faults like images not only vibrations. You have to mention why you chose audio signals.
Why did not you consider lightweight pre-trained deep learning models?
Introduction section is too long. Could you please move the literature to another section.
Method and Materials
Why did you transform signals to gray mages not RGB?
Please justify the size of segments.
Please add more details regarding the dataset.
Did you use any augmentation techniques?
Experimental Results
Please define performance metrics used and add their equations.
Other performance metrics should be added like sensitivity, specificity, precision, and F1-score.
The conclusion
Please mention the limitations of your technique and your future work.
Round 2
Reviewer 1 Report
The current version has addressed all my issues and there are no further comments.
no
Author Response
Thanks for the reviewer's affirmation.
Reviewer 3 Report
Thank you authors for addressing all the review comments.
Author Response
Thanks for the reviewer's affirmation.
Reviewer 4 Report
I cannot find the justification of segmented window size. please state clearly where and what changes have been made to the manuscript.
in lines 269-272, please replace "datasets" of the data.
I asked the authors if they used augmentation techniques but their response is irrelevant and they did not addressed this issue
